# Changes in the Impacts of COVID-19 over Time on Families with Older Adults Living on Remote Islands in Japan: A Study in Family Ethnographic Research

**DOI:** 10.3390/healthcare11233088

**Published:** 2023-12-02

**Authors:** Naohiro Hohashi, Mikio Watanabe, Minami Taniguchi, Shiho Araki

**Affiliations:** 1Division of Family Health Care Nursing, Graduate School of Health Sciences, Kobe University, Kobe 654-0142, Japan; 2Department of Nursing, Faculty of Health Sciences, School of Medicine, Kobe University, Kobe 654-0142, Japan

**Keywords:** COVID-19, remote island, family with older adults, Concentric Sphere Family Environment Theory, family ethnographic research

## Abstract

Coronavirus disease 2019 (COVID-19) affected not only individuals but also families. The purpose of this study was to clarify the temporal changes in the impact of the COVID-19 pandemic on entire families with older adults susceptible to infection living on small islands in Japan over the duration of the pandemic. Family ethnographic research was conducted from 2021 to 2023, using the Concentric Sphere Family Environment Theory as the theoretical framework. Formal interviews were conducted with 20 families. In addition, data from informal interviews, participant observation and other sources were compiled into field notes. All data on the impact on the entire family were extracted and content analysis was conducted. Six categories (family internal environmental system, family system unit, micro system, macro system, supra system, and family chrono-environment system) and a total of 85 subcategories were extracted. The results show that COVID-19 exerted not only negative but also positive impacts on the entire family, and their temporal changes are clarified. The impact on families is believed to have been influenced by the family external environment, such as increases and decreases of infection cases or events that occurred outside the family. The knowledge acquired from these studies will help healthcare professionals in providing appropriate family support.

## 1. Introduction

The first cases of novel coronavirus infection were reported in Japan on 16 January 2020 [1]. The Japanese government declared a state of emergency and asked the public to refrain from going out unnecessarily, but the lockdown was voluntary, and the government did not have the power to compel the public as was the case in other countries [2]. The coronavirus disease 2019 (COVID-19) pandemic, however, has been found to have exerted a negative impact on the mental health not only of individuals [3], but also on the entire family. For example, negative impacts, such as worsening of family relationships and deterioration of family functions, have been reported [4].

Older adults are at higher risk than others for serious COVID-19 outcomes, including hospitalization and death, and social distancing was paramount to prevent the spread of COVID-19 in older adults [5]. However, social distancing may lead to inactivity and social isolation, and the impact of COVID-19 among family members, particularly older adults, may have been especially strong. Older adults tend to maintain a wide social range, which might include not only relatives, but also neighbors, religious practices and interactions with other local communities, with which longstanding and deep relationships exist [6,7]. This suggests that the impact of COVID-19 was widely felt in ties between the internal and external family environments of older adults.

Following the outbreak of COVID-19, the number of newly infected cases repeatedly increased and decreased, and associated impacts are considered to have occurred over time [8]. In Japan, the first wave began on 29 January 2020, and the number of cases increased rapidly in the seventh wave beginning from 25 June 2022; the COVID-19 pandemic may have had a different impact, or the impact may have changed, over time since the initial outbreak [9]. Therefore, it is necessary to clarify the impact of the COVID-19 pandemic on families over time. The impact of COVID-19 on families is considered to vary not only with the internal environment of the family, such as the occupations of family members and family composition, but also with the external environment of the family, such as policies regarding behavioral restrictions and school closures [10]. Therefore, it is necessary to examine a broad perspective of the environment surrounding the family while clarifying the impact on the family.

The Concentric Sphere Family Environment Theory (CSFET) proposed by Hohashi [11] is a middle-range family nursing theory that takes a holistic view of the family environment that affects the well-being of the family system unit [12]. CSFET is composed of five systems: (1) family internal environment system; (2) supra system; (3) macro system; (4) micro system; and (5) chrono system. The supra, macro, and micro systems belong to the family external environment system. Utilizing CSFET, it is possible to comprehensively capture the impact of COVID-19 on both the family and its external environment. The chrono system is also expected to reveal the impact of COVID-19 over time. Therefore, we adopted CSFET as the theoretical framework for this study.

The Japanese archipelago consists of four main islands, with 14,125 islands in total [13]. Due to the geographical characteristics of being surrounded by ocean, some islands face difficulties in transportation access which has led to the stagnation of industrial activities, a subsistence lifestyle, and inadequate education, medical care, and welfare systems. This is especially so in terms of population decline and an aging society with low fertility [14]. In addition, because few people come and go, the islanders maintain a mutually recognizable relationship, and a distinctive traditional culture remains [15]. The environment on such islands is believed to be unique in terms of the impact of COVID-19.

In this study, the CSFET was utilized as a theoretical framework to clarify the temporal changes in the impact of the COVID-19 pandemic on families with older adults living on an island during the periods of increased infection (from the first through the seventh wave).

## 2. Methods

### 2.1. Research Design

This is a qualitative descriptive study employing family ethnographic research [16] to share experiences in family life settings. In order to investigate the impact of COVID-19 on families living on the islands based on their culture, values, and island characteristics, the researcher personally entered and lived with the islanders in their living environment. Long-term family ethnographic research aimed to find a holistic view of the various influences on the family. The research methods included participant observation, ethnographic interviews (formal and informal interviews), collection of existing books, literature, and Internet resources, and photography [16], with CSFET (Figure 1) as the theoretical framework [11,12].

The operational definition and meaning of the terms used are as follows:Impact: “The result of an action by one thing on another” [17]. Impact has a direction (positive or negative) and a magnitude (amount of change). A primary impact may be accompanied by another impact (secondary impact), and such a chain of impacts (secondary impact, tertiary impact, etc.) is also included in the generic term “impact”.Periods of increased infection (waves): 1st wave, from 29 January 2020 to 13 June 2020; 2nd wave, from 14 June 2020 to 9 October 2020; 3rd wave, from 10 October 2020 to 28 February 2021; 4th wave, from 1 March 2021 to 20 June 2021; 5th wave, 21 June 2021 to 16 December 2021; 6th wave, from 17 December 2021 to 24 June 2022; and 7th wave, 25 June to 26 September 2022 [18].Family: “A unit/organization as a system of the OR operation (logical operation) of individuals, that is, living people, having the cognition of belonging by other constituent member(s)” [19].Family internal environment system: “The environment that includes family communications ability; family time allocation; beliefs of family/family members; family health-related self-care ability; family economic power; family living environment and others” [12].Family system unit: “A family existing as a system and a unit” [12].Family external environment system: “The 3 environments consisting of supra system, macro system, and micro system” [12].Micro system: “The environments that include the local living sphere; relatives, family friends; neighbors and others” [12].Macro system: “The environments that include educational, nursery, adult learning facilities; health, medical, welfare facilities; workplace environment; social resources, public services; politics, economics and others” [12].Supra system: “The environments that include religion; culture; the supernatural and superscientific cosmos and others” [12].Family chrono-environment system: “The environments that include adapting to family events; family chronicle; realizing family demands/family hopes and others” [12].

### 2.2. Participants and Setting

The field of the family ethnographic research was four islands belonging to a municipality located on Japanese islands, and the participants were families with older adults aged 65 or older. The municipality consists of 10 inhabited and 53 uninhabited islands with a population of 34,391, of which 14,002 (40.7% of population) are aged 65 or older [20]. The islands are famous for fishing, agriculture, and tourism. The first case of novel coronavirus infection on the island was confirmed on 20 July 2020.

For formal interviews, we explained the study objectives, methods, protection of personal data, the inclusion criteria for participants, and others were explained at five facilities (a visiting nursing station, a visiting home care station, a welfare facility for the older adults, a social welfare council, and a city council) and to three gatekeepers. After the families with older adults were introduced, the researcher contacted the families to obtain their consent to participate in this study. The participating family member(s) including older adult(s) had to be willing to participate in a 120 min interview and reply to the field worker’s questions. In order to comprehensively study a variety of families, such specifics as the older adult’s level of care, illness, and family structure were not included in the selection criteria.

Formal interviews were conducted with a gradually increasing number of family, eventually reaching 20 families (34 family members). The average interview time was 105 min (range, 58 to 135 min). The mean number of family members was 5.5 (range: 2–19). The mean age of the older adults was 74.9 years old (range: 65–97). The most common family medical history conditions were hypertension (6 family members), diabetes (3 family members) and malignant neoplasms (2 family members). Professed religious beliefs were Buddhist (12 families), nonreligious (7 families) and Christian (1 family). All families had family members living off-island.

### 2.3. Data Collection

From January 2021 to June 2023, researchers entered the island and conducted fieldwork for a total of 230 days. During the observation and informal interviews, a field notebook was always in hand to write down in detail what was seen and heard, impressions, the general atmosphere, and others. Photographs were taken when necessary. During the informal and formal interviews, nonverbal communication such as the participant’s facial expressions and the researcher’s impressions were also recorded in the field notebook.

From January 2021 to March 2022, books, literature, and Internet resources were collected on the islands and their impact on the lives of the older adults. Specifically, we collected articles using literature databases (Ichushi-Web, which is a database of articles published in Japanese medical journals, and Google Scholar), and continuously collected local histories and statistical data at the municipal library on the islands, and all available resources.

From July 2022 to March 2023, semi-structured interviews were conducted with each family as formal interviews based on the culture and values of the families living on the islands and the characteristics of the islands. The Family Environment Assessment Index (FEAI) [21], a collection of sample questions to systematically and accurately collect family information necessary for family assessment, was used as a guide for the interview, which was appropriate for the purpose of this study. The interview guide was developed for the purpose of this study. Formal interviews were conducted with three families who cooperated as a convenience sample, the interview guide was improved by adding explanations to focus on the impact on the family as a whole, rather than on individual family members, and others. In addition, in order to effectively conduct the formal interviews by the periods of increased infection [22], we created a diagram (Figure 2) that summarized the main events for each period of infection spread, and conducted the interviews while checking the period of infection spread during which the impact on families occurred. Examples from the interview guide include “Was there anything with which the whole family had difficulty?” “Was there anything that helped the whole family?” “Did anything change in family relationships?” “Has the significance of the family’s existence changed at any time?” “Has the change or influence of the family had any further impact on your family? And if so, why?” Family attributes such as family structure were collected using the Family Environment Map (FEM) [23]. The formal interviews lasted approximately 120 min per family, and the conversations were recorded with an IC recorder.

### 2.4. Data Analysis

A verbatim transcript of the formal interviews was prepared from the recordings. Interpretations of the contents of photographs and information obtained from books, literature, and Internet resources were added to the field notes. That is, all data of the ethnographic research were described in field notes. All of these data were then repeatedly read back to understand the overall context. The impact of the COVID-19 pandemic on the families was extracted by classifying them into periods of increased infection (from the first through the seventh wave), and directed content analysis was conducted based on CSFET [24]. For details, the smallest semantic unit was coded, and subcategories and categories (the five systems of CSFET) were created from codes with the same semantic content [25].

### 2.5. Trustworthiness

Formal interviews were conducted by a researcher skilled in family ethnographic research. Coding and analysis were conducted at the end of each interview and at the end of daily fieldwork using the constant comparative method. Formal interviews were terminated after 20 families had been interviewed, because the same responses were obtained from which no new information was added and data were considered saturated. In subcategorization and categorization, subcategory and category names were repeatedly examined until the four researchers reached a consensus, and the same recognition was achieved among the researchers. The results of the analysis were returned to the target families of the formal interviews, and family member checks [26] were conducted. In addition, the analysis was reviewed with nine experts in family nursing and qualitative research in order to ensure trustworthiness.

### 2.6. Ethical Consideration

This study was conducted after obtaining the approval of the affiliated university’s Institutional Review Board. For all members of all the participating families in the formal interviews, the study objectives, methods, reasons for audio recording, right to withdraw, and protection of personal data were explained, with permission obtained in writing. During the fieldwork, the protection of personal data and the right to withdraw were verbally explained to the participants. Both participants and field workers continuously adopted thorough precautions to prevent COVID-19 transmission.

## 3. Results

Table 1 shows the positive impact and Table 2 shows the negative impact of COVID-19 on families over the duration that the infection spread.

### 3.1. Impact of the First Wave

Positive impacts on families were observed in the family system unit, “Fostering a feeling of mutually helping one another as a family” (P1), and in the supra system, “Due to the death of a prominent person from COVID-19, families’ becoming more aware of infection control measures” (P7). On the other hand, negative impacts were observed in the micro system, “Disappearance of opportunities for interaction with neighbors and family” (N5), and in the family chrono-environment system, “Family anxiety about the unknown aspects of COVID-19” (N6). In the first wave, because scant information about COVID-19 existed and it was an unknown virus, one formal interviewee, for example, remarked “We didn’t know what kind of disease it was, so we were very anxious. We were worried about what would happen if we were to develop some kind of mental problem in our lives.”.

### 3.2. Impact of the Second Wave

Positive impacts on families were observed in the family system unit, “Shifting the family’s thinking to a more positive orientation about restrictions on activities” (P10), and in the macro system, “Development of a sense of crisis regarding COVID-19 through consideration of the island’s characteristics” (P13). On the other hand, negative impacts were observed in the family system unit, “Family’s fear of becoming a source of infection on the island” (N11), and in the macro system, “Increased fear of infection caused by rumors concerning the first person to be infected on the island” (N15). The first confirmed cases in the small island were in the second wave, and rumors led to a change in the family’s awareness. For example, one formal interviewee remarked: “At first, two of the islanders went off-island and got infected. The rumors were quite serious. We heard such stories over and over again. So we knew that if we mistakenly contracted COVID-19, we would be in big trouble, and we didn’t know what everyone would say.”.

### 3.3. Impact of the Third Wave

Positive impacts on families were observed, such in the family system unit, “Family’s secure feelings about infection prevention with the advent of vaccines” (P18), and in the supra system, “Strengthening mutual help between people and families on the island through a religious-based spirit of mutual help” (P23). On the other hand, negative impacts were observed in the family system unit, “Differences of opinion within the family regarding the pros and cons of vaccination” (N20), and in the supra system, “Family spiritual suffering due to cancellation of religious events” (N22). In the third wave, both positive and negative impacts were observed due to the introduction of vaccines. For example, one formal interviewee remarked “I was afraid to be vaccinated because I didn’t feel like getting vaccinated, but my wife insisted that I get the vaccine, so I had no choice.”.

### 3.4. Impact of the Fourth Wave

Positive impacts on families were found in the macro system, “Diminishing of fear of infection through gathering of information” (P28), and in the family chrono-environment system, “Deeper consideration concerning lifespans of older family members” (P29). On the other hand, negative impacts were observed in the macro system, “Dissatisfaction with lack of information via disaster prevention radio” (N26), and in the family chrono-environment system, “Anxiety about vaccination side effects” (N29). In the fourth wave, information about and the characteristics of COVID-19 gradually increased, and changes in the image of COVID-19 were observed. For example, one formal interviewee remarked “Well, rumors are just that, rumors. As for the anxiety, what do I do? I just gather information from the local news, right? It must be the accurate information.”.

### 3.5. Impact of the Fifth Wave

Positive impacts on families were found in the family system unit, “Family’s closer feelings towards family members who have become a close contact of a case of coronavirus” (P33), and in the supra system, “Generation of positive feelings among families by holding a global festival” (P36). On the other hand, negative impacts on families were observed, such as, in the family system unit, “Occurrence of family acclimatization to infection due to infection of family members” (N32), and, in the supra system, “Sharp differences in awareness of the risk of infection between generations” (N34). Even among family members, the awareness of the risk of infection differed by generation. For example, one formal interviewee remarked “Our father is older, and he is not well informed because he spends most of his time in relationships with people on a closed, remote island. Therefore, he is less aware than we are of infection prevention behaviors.”.

### 3.6. Impact of the Sixth Wave

Positive impacts on families were found in the family system unit, “Recovery of family economy thanks to increase in number of tourists” (P38), and in the micro system, “Easing neighbors’ views toward people leaving the island by implementing antigen tests” (P39). On the other hand, negative impacts on family members were observed, such as, in the family internal environmental system, “Reduced sense of crisis due to asymptomatic infection” (N35), and, in the family chrono-environment system, “Concerns over infection caused by at-home treatment of infected family members” (N37). As the island’s economy is also focused on tourism, one formal interviewee, for example, made this remark concerning the increase in the number of tourists: “When the number of tourists declined, we were wondering what would happen to our own business, but during the vacations when there were no restrictions, lots of tourists came, and merchandise sold well. Thanks to everyone, we were very happy.”.

### 3.7. Impact of the Seventh Wave

Positive impacts on families were found in the family system unit, “Experiencing family’s happiness from being reunited with family members who returned to the island from off-island” (P42), and in the micro system, “Relief from stress caused by concerns for neighbors” (P43). On the other hand, negative impacts on family members were observed, such as, in the family system unit, “Increased anxiety of families due to the rapid increase in COVID-19 infections on the island” (N38), and, in the family chrono-environment system, “Resignation toward becoming infected as the number of infections is increasing” (N40). In the seventh wave, it became realistic to continue living with COVID-19, and one formal interviewee, for example, said “We have come to think that we are going to have to live with the coronas for a long time, so we have given up. We have no choice but to do the same thing we did in the first wave, or I just do what we can do.”.

## 4. Discussion

### 4.1. Overall Picture of the Impact of COVID-19 on Families

In the closed and distinctive family environment existing on a Japanese island, a long-term family ethnographic study [16] of families with older adults susceptible to COVID-19 infection provided rich data on the impact of COVID-19 on these families. Using CSFET as a theoretical framework, we were able to obtain a comprehensive understanding of the impact of COVID-19, not only within the family but also outside of the family. This is because the family is a system and a unit that interacts constantly with the outside environment (relatives, neighbors, work/school, hospital, politics, economy, etc.) [17], and the application of CSFET to this study proved effective [12]. In particular, the number of infected people on the island and the policy for COVID-19 were the causes of the impact, and the impact on families was changing with the change in these over time, so it is important for nursing professionals to be involved in the policy [27].

The impact of COVID-19 on families was characterized by both negative and positive aspects. This may be due, in part, to the unprecedented family event of COVID-19, which resulted in the heightening of family resilience [17]. Family resilience, in family nursing, is demonstrated “when a family becomes aware of family symptoms, its power to autonomously and actively improve its own family functions” [17]. Family resilience, therefore, is a key factor in the maintenance and improvement of family relationships and family functions in the event of a family crisis. Therefore, preventive family support to enhance family resilience is essential to prepare for sudden outbreaks of emerging infectious disease in the future [28].

The impact on families is believed to have been influenced by the family external environment, such as during periods of increased infection (increases and decreases in infection cases) or events that occurred outside the family (the death of a prominent individual due to COVID-19, news about the vaccine, etc.). The subcategories for each period of increased infection up to the seventh wave, when the number of infections increased rapidly, indicate that family attitudes changed over the passage of time. Because changes in family functioning have been observed within a week after a family event [19], it is possible that family events occurring outside of the family may also foster changes in the impact.

In addition to taking into account the characteristics of the family environment, as well as the individuality of the family, nursing professionals are obliged to conduct holistic family assessments based on the CSFET, taking into account the family’s internal, external, and temporal factors, in order to provide appropriate family support. In the following, we discuss the positive and negative impacts, and changes over time, based on the subcategories identified in this study.

### 4.2. Positive and Negative Impacts of COVID-19 on the Families and Their Changes

In the first wave, very little information on COVID-19 was available, and uncertainties about the identity of the virus caused considerable anxiety among families. In particular, the deaths of prominent Japanese people due to COVID-19 appeared successively in the news (Figure 2), and we believe this led to the strengthening of family awareness to treat infection control measures as a threat. In Japan, a policy of refraining from going out was announced but no mandatory lockdown was implemented. Nevertheless, Japanese people began to stay home voluntarily. The family is the entity that can be relied on in such a crisis situation, and we believe that this fostered a sense of mutual support within the family. Although the term “corona divorce” attracted some attention on the Internet, the divorce rate in Japan conversely decreased, suggesting that family ties were strengthened due to economic uncertainties over the future and others [29]. The self-imposed restraints against going out eliminated opportunities for family relations with neighbors, and this is particularly associated with anxiety, depression, poor sleep quality, and physical inactivity among the older adults [30], so consideration for the families with older adults was necessary.

Because COVID-19 was not transmitted from outside the island to the island during the first wave, no cases of infection occurred, and we believe that the family’s fear and sense of danger of becoming the first infected person on the island emerged in the second wave, leading to a more positive acceptance of behavioral restrictions and a shift to a more positive attitude by the family. In the first wave, the family members, who had been fully occupied with self-help within the family, began to look outside the family. In the second wave, the first cases of infection were confirmed on the island, and in fact, personal and discriminatory information in the form of rumors about the first cases spread throughout the island. This suggests that it is difficult to maintain anonymity [31] on islands, even if information remains highly anonymous in urban areas, and that it is necessary to tailor responses according to regional characteristics.

The advent of the vaccine was a major family event in the third wave, and vaccinations targeting healthcare workers were initiated around the end of the third wave (Figure 2). Even when vaccinations had not yet begun for ordinarily families, the vaccine was perceived as a highly effective infection prevention measure, and likely resulted in a sense of security for the family. On the other hand, the fact that the vaccine was a hastily developed mRNA vaccine, the effectiveness of which was unknown, and the fact that the number of infected persons was increasing despite the progress of vaccination, were thought to have heightened family anxiety and fear [32,33]. This chaotic situation may have led to conflicting opinions within families regarding the pros and cons of vaccination. The third wave overlapped with the “Oshogatsu” New Year’s holiday, an important event for Japanese people when they welcome the New Year and New Year’s deities. Despite it being a public holiday in Japan and lasting over an extended period, religious events at this time were canceled, resulting in spiritual suffering in the family and a need for religious family support [34].

On the other hand, in the fourth wave, family fears of infection due to information about COVID-19 decreased, suggesting that accurate information was being disseminated [35]. However, misinformation about vaccine side effects was spread via social media, causing families to be concerned about post-vaccination side effects. Since obtaining health and medical information has the significant effect of providing a sense of security, families with high health literacy were more likely to have seen a decrease in fear of infection [36]. Because rumors tend to spread easily on islands, it is important to provide correct information. On islands, disaster radio broadcasting serves as an important means of obtaining information, and it is important to quickly provide accurate information to family members. The disaster prevention radio system is a push-type information delivery system from the government, and while it is important for families to actively collect information on their own, it is also important for the government to disseminate information in a variety of ways. Furthermore, in the fourth wave, there was a deepening of consideration for the life expectancy of older adult family members as they refrained from going out for extended periods. This is something that affects views of life and death, and we believe that the spirit of mutual aid through religion has been strengthened among the people and their families on the island. Just as religious services were cancelled in the third wave, spiritual care for family members took on greater importance [37].

In Japan, the number of cases began to increase in the fifth wave, which was marked by the emergence of the Omicron strain, and the number of infected family members on the island also increased, suggesting that family members had become accustomed to the idea of becoming infected. However, older parents were generally more susceptible to infection and more aware of strict infection control measures. On the other hand, family members (offspring of the older individuals), who generally have a higher resistance to infection, are less conscious of infection control measures, and differences in infection risk awareness between generations became apparent [38]. As the number of infected people began to increase, the use of contact confirmation applications utilizing smart phones began to increase, and the number of those who had close contact with people infected with the novel coronavirus increased. When a family member becomes a close contact, he or she must be treated at home, forcing the family to unite. In addition, Japan hosted the 2020 Summer Olympic and Paralympic Games, which were postponed for one year, and the sight of athletes competing in the midst of the COVID-19 pandemic [39] brought encouragement to family, which is thought to have generated positive feelings.

In the sixth wave, the number of tourists, with tourism being one of the most important industries on the island, began to recover, and families engaged in tourism-related occupations showed a recovery in family economic strength. Since the largest influencing factor of family functioning is family economic strength [19], we believe that the recovery of tourism led to the recovery of family well-being. Although collaboration among other professions in nursing is said to be important, we believe that it is necessary to collaborate not only with health, medical care, and welfare, but also with all professions related to family life [40]. We also believe that, in the sixth wave, the easy availability of free antigen testing eased the neighbors’ view toward going outside the island, and that antigen testing has made it easier for people to go outside the island. On the other hand, inapparent infection began to occur, and the family’s sense of risk toward COVID-19 decreased. However, family members with apparent infection were compelled to receive treatment at home due to a shortage of beds at medical institutions, and family anxiety about infection from infected family members increased. We believe that families developed various coping strategies against COVID-19 as a result of the prolonged pandemic.

In the 7th Wave, the government’s recommendation to refrain from leaving home was lifted, and more family members living outside the island returned and felt the joy of being reunited with their families. The islanders were able to go out without hesitation, and they did not have to worry about what their neighbors would think, which provided relief from stress. However, the number of people infected with the novel coronavirus on the island continued to increase rapidly, and the family’s anxiety increased because it appeared that anyone could be infected at any time. In an island where medical care is tenuous [41], older individuals are especially vulnerable to infection and their infection leads to anxiety. Thus, according to the number of infected persons, family members’ emotions swung, with feelings of relief when the number of infected persons on the island was low and feelings of anxiety when the number was high. On the other hand, the explosive increase in the number of cases of infection also led to resignation among family members that they would become infected. This may have been due to stress caused by the long-term COVID-19 pandemic, and it was evident that the family members were conflicted with ambivalent feelings. At the same time, it was thought that their attitudes toward coexistence with COVID-19 were undergoing change [42].

### 4.3. Limitations and Future Directions

This study accepts that it has certain limitations. Because it incorporates data obtained by recalling past impacts on families, recall bias may exist. Additionally, during the formal interviews, not all family members, including older adults, were able to participate, so the impact on the entire family may not have been discussed. In the future, it will be desirable to continue this research after the COVID-19 pandemic has ended, to evaluate the long-term impact on families.

## 5. Conclusions

In this study, under the crisis situation of the COVID-19 pandemic, we targeted families with older adults with weakened immune systems living on a small remote island in Japan, an isolated area distinguished by few people coming and going. This kind of nursing research has significance as a historical record. Through long-term family ethnographic research, we have been able to clarify that the pandemic resulted not only in negative but also positive impacts on entire families, and detailed how changes in these impacts occurred over time, and took into account the need for timely family support and consideration of family distinctions. The adoption of CSFET, a middle-range family nursing theory, as a theoretical framework made it possible to understand the family as a system, and to study influences on the whole family through interactions/transactions not only with the family internal environment but also the family external environment, and we were able to reconfirm the importance of family support from a holistic perspective. Family is one of the most important things for people, and support for the entire family is essential for improving the well-being of individual family members and of the entire family.

## Figures and Tables

**Figure 1 healthcare-11-03088-f001:**
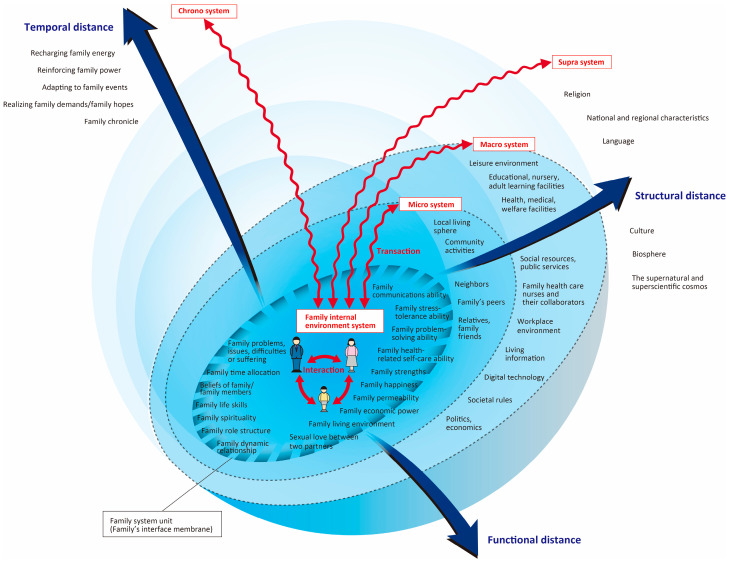
Model diagram of Concentric Sphere Family Environment Theory (CSFET) (Ver. 3.4).

**Figure 2 healthcare-11-03088-f002:**
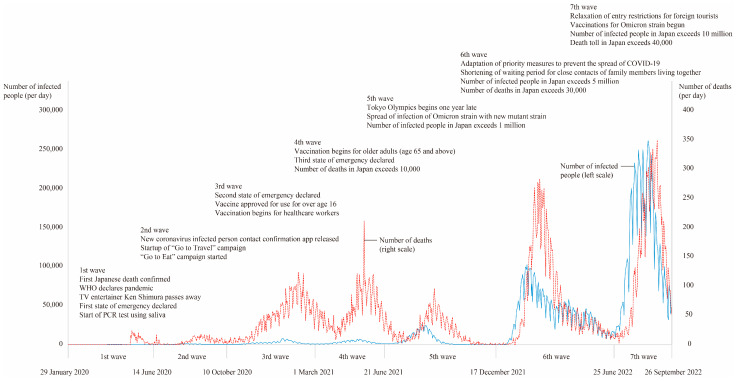
Chronology of Japan’s COVID-19 situation and related events by wave, 2020–2022. Note: red line = number of deaths (right scale); blue line = number of infected people (left scale).

**Table 1 healthcare-11-03088-t001:** Positive impact by wave of COVID-19 on families with older adults living on remote islands.

Category (System)	Subcategory						
	1st Wave	2nd Wave	3rd Wave	4th Wave	5th Wave	6th Wave	7th Wave
Int		(P9) Improving awareness of health-related activities, including infection prevention among family members	(P17) Strengthen awareness of infection prevention among family members as the island lacks good medical facilities		(P31) Psychological stability of family members enabled by information exchange between family members		(P41) Despite the increase in the number of infected people, resumption of return of family members living outside the island thanks to dropping of restrictions
Fsu	(P1) Fostering a feeling of mutually helping one another as a family (P2) Strengthening family bonds living on the same islands by discouraging the return of family members living off-island (P3) Reduced burden on household finances due to restrictions on leisure activities	(P10) Shifting the family’s thinking to a more positive orientation about restrictions on activities (P11) Appearance of a family belief that health comes first (P12) Improvement in family self-care skills through increased frequency of contact between family members	(P18) Family’s secure feelings about infection prevention with the advent of vaccines (P19) Family cooperation to alleviate loneliness from restrictions toward family members living off-island from returning	(P24) Reducing family’s concerns over the risk of infection through vaccination and testing (P25) Family’s relief that no one became infected at a funeral	(P32) Family’s sense of security concerning family members avoiding infection through additional vaccinations (P33) Family’s closer feelings towards family members who have become a close contact of a case of coronavirus	(P38) Recovery of family economy thanks to increase in number of tourists	(P42) Experiencing family’s happiness from being reunited with family members who returned to the island from outside the island
Mic	(P4) Maintaining a sense of community solidarity despite the cancellation of local events		(P20) Providing support to relatives using SNS	(P26) Family promoting infection-prevention behavior to other family and friends		(P39) Easing neighbors’ views toward people leaving the island by implementing antigen tests	(P43) Relief from stress caused by concerns for neighbors
Mac	(P5) Strengthening the sense of norms regarding family respectability	(P13) Development of a sense of crisis regarding COVID-19 through consideration of the island’s characteristics (P14) Improvement of family members’ sense of crisis regarding infectious diseases by policies	(P21) Strengthening family awareness of infection prevention through information gathering (P22) Request for disclosure of details such as the district name where infections occurred	(P27) Sense of security for movement to and from the island through vaccination and testing (P28) Diminishing of fear of infection through gathering of information	(P34) Utilization of disaster prevention radio as a means of obtaining information (P35) Increased sense of security among family members toward island visitors who have undergone antigen testing	(P40) Family members providing mutual help via SNS when a family member living off the island became infected	(P44) Closer psychological proximity with island visitors
Sup	(P6) Increased free time among family members due to business closures (P7) Due to the death of a prominent person from COVID-19, families became more aware of infection control measures	(P15) Improving awareness of infection prevention through changes in the ways of holding funerals	(P23) Strengthening mutual help between people and families on the island through a religious-based spirit of mutual help		(P36) Generation of positive feelings among families by holding a global festival		
Chr	(P8) Increase awareness of the presence of families living on the island	(P16) Increase in time with family thanks to more opportunities to go out together		(P29) Deeper consideration concerning lifespans of older family members (P30) Changes in the practice of visits to family graves based on infection status	(P37) Hope for reunion with family members living off the island thanks to the ban on homecoming		(P45) Lives not wishing to change infection control measures even after the pandemic

Note: Int = family internal environmental system; Fsu = family system unit; Mic = micro system; Mac = macro system; Sup = supra system; Chr = family chrono-environment system; 1st wave = from 29 January 2020 to 13 June 2020; 2nd wave = from 14 June 2020 to 9 October 2020; 3rd wave = from 10 October 2020 to 28 February 2021; 4th wave = from 1 March 2021 to 20 June 2021; 5th wave = from 21 June 2021 to 16 December 2021; 6th wave = from 17 December 2021 to 24 June 2022; 7th wave = from 25 June 2022 to 26 September 2022.

**Table 2 healthcare-11-03088-t002:** Negative impact by wave of COVID-19 on families with older adults living on remote islands.

Category (System)	Subcategory						
	1st Wave	2nd Wave	3rd Wave	4th Wave	5th Wave	6th Wave	7th Wave
Int	(N1) Fear of infection among family members with illnesses/disabilities or older adults (N2) Loss of opportunities to meet family members who live apart	(N7) Restrictions on work and facility use for family members after their return to the island (N8) Psychological burden on family members due to changes in work		(N23) Difficulty in receiving vaccinations for older family members on their own	(N30) Difficulty in maintaining distance from close contacts who are quarantined at home (N31) Increased fear of side effects from vaccine	(N35) Reduced sense of crisis due to asymptomatic infection	
Fsu	(N3) Imbalance in family members’ roles due to lack of change in family role apportionment (N4) Family concerns about members living outside the island	(N9) Decrease in family income due to drop-off in number of tourists (N10) Occurrence of loneliness caused by cancellation of family events (N11) Family’s fear of becoming a source of infection on the island	(N20) Differences of opinion within the family regarding the pros and cons of vaccination	(N24) Family distress over infection prevention for funeral attendees from outside the island	(N32) Occurrence of family acclimatization to infection due to infection of family members		(N38) Increased anxiety of families due to the rapid increase in COVID-19 infections on the island
Mic	(N5) Disappearance of opportunities for interaction with neighbors and family	(N12) Negative comments from neighbors about an off-island trip (N13) Discriminatory treatment due to suspicion of infection from neighbors after returning to the island		(N25) Decrease in daily mutual assistance between neighbors	(N33) Increase in rumors about infected people in the neighborhood		
Mac		(N14) Decrease in family happiness due to loss of opportunity for family members living away from the island to return (N15) Increased fear of infection caused by rumors concerning the first person to be infected on the island (N16) Difficulty in obtaining face masks due to rising prices stemming from a nationwide mask shortage	(N21) Difficulty meeting grandchildren born outside the island for the first time	(N26) Dissatisfaction with lack of information via disaster prevention radio (N27) Family concerns over lack of medical facilities on the island (N28) Decreased sense of urgency due to lack infected people in proximity			(N39) Expectations that payment of benefits will exacerbate the decline in economic strength
Sup		(N17) Psychological instability within family due to cancellations of local and traditional events (N18) Cancellation of family members living outside the island to return and participate in religious events	(N22) Family spiritual suffering due to cancellation of religious events		(N34) Sharp differences in awareness of the risk of infection between generations	(N36) Increased sense of crisis due to news of exploding rate of infection	
Chr	(N6) Family anxiety about the unknown aspects of COVID-19	(N19) Family anxiety about inability to adopt ‘perfect’ infection prevention measures		(N29) Anxiety about vaccination side effects		(N37) Concerns over infection caused by at-home treatment of infected family members	(N40) Resignation toward becoming infected as the number of infections is increasing

Note: Int = family internal environmental system; Fsu = family system unit; Mic = micro system; Mac = macro system; Sup = supra system; Chr = family chrono-environment system; 1st wave = from 29 January 2020 to 13 June 2020; 2nd wave = from 14 June 2020 to 9 October 2020; 3rd wave = from 10 October 2020 to 28 February 2021; 4th wave = from 1 March 2021 to 20 June 2021; 5th wave = from 21 June 2021 to 16 December 2021; 6th wave = from 17 December 2021 to 24 June 2022; 7th wave = from 25 June 2022 to 26 September 2022.

## Data Availability

The data are not publicly available for reasons of privacy or due to ethical restrictions.

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
