# Peer review of "Changes in the Impacts of COVID-19 over Time on Families with Older Adults Living on Remote Islands in Japan: A Study in Family Ethnographic Research"

_healthcare, 2023, doi:10.3390/healthcare11233088_

Round 1
Reviewer 1 Report
Comments and Suggestions for Authors
The topic is very interesting when dealing with Changes in the impacts of COVID-19 over time on families with older adults. Not only the negative aspects are addressed, but the positive aspects that the pandemic also brought are taken into account. The way of explaining it, the methodology and the order of presentation are appropriate, I would just like to add some information:
1. In the abstract add the range of members per family, the average age of older adults and specify the 5 categories studied.
2. Specify the starting hypothesis, that is, what the researchers hope to find.
3. Add to the participants section, if it has been asked, the sex, level of education, employment status,... of the families. If a table could be added with all this information it would be much more visual.
Good job.
Author Response
Dear Reviewer 1:
Thank you for your comments. In response we have taken the following measures:
Your comment:
1. In the abstract add the range of members per family, the average age of older adults and specify the 5 categories studied.
Our answer:
Since the number of categories was mistakenly given as 5, but was actually 6. This has been corrected. The data considering of formal interviews was only part of the entire data, so this is not believed to be important. However, as you pointed out, the number of words in the abstract were over the permitted amount. From line 18 to line 20 we have added 6 categories. Elsewhere, this was added from line 144 to line 145.
Your comment:
2. Specify the starting hypothesis, that is, what the researchers hope to find.
Our answer:
This study was ethnographic research, and as it is not quantitative research, a starting hypothesis is unnecessary. As you pointed out, however, we have added a brief note as an aim to lines 86-87.
Your comment:
3. Add to the participants section, if it has been asked, the sex, level of education, employment status,... of the families. If a table could be added with all this information it would be much more visual.
Our answer:
The study was ethnographic research, so the data from the formal interviews was only part of the entire data. In semi-structured interviews, we inquired about a various of family characteristics such as family composition and so on. In 1, above, we have added members per family and age of older adults to the main text, and determined this should be sufficient.
Reviewer 2 Report
Comments and Suggestions for Authors
Thank you for the opportunity to review this interesting article.
This qualitative descriptive study is clear, well-written and interesting. I have made only a couple of comments, asked some questions that need consideration, and given suggestions for change/improvement in some cases.
Background/Lit Review
This section is well-written. The study focuses on older adults 65y years or more.
Methods
This entire section is well covered and explained in the study. All the relevant parts of the study were explained.
Design. The 11 purpose of this study was to clarify the temporal changes in the impact of the COVID-19 pandemic 12 on entire families with older adults susceptible to infection living on small islands in Japan over the 7 waves of the pandemic experienced in Japan. The theoretical framework used was also well explained and the figure/visual model provided which is helpful for readers.
Participants. A statement regarding IRB review/approval and informed consent process is included and the process well explained for seeking the approval and informed consent.
- Did younger people/family members play any role in this study? Was the role of younger people considered? Why or why not?
Data Collection. The research methods included participant observation, ethnographic interviews 84 (formal and informal interviews), collection of existing books, literature, and Internet re- 85 sources, and photography.
The use of different methods of data collection helps to strengthen the type and quality of data collected. This adds to the inter-rater reliability check and reaching consensus on themes and key findings/outcomes in relation to the purpose of the study.
Data Analysis. This is a very thorough description of your process to assess the impact of COVID-19 within the various levels in the Concentric Sphere Family Environment Theory (CSFET)/framework.
However, the main highlight was on ‘content analysis’. I am not sure that stating “content analysis for this study is a right term’. What was done for this analysis as I understand it is more of “thematic analysis’ and not content analysis’. The authors should check on this and make necessary adjustments.
Rigor. This is explained on how trustworthiness was attained. However, since different sources of information were utilized for data collection, it is not clear how triangulation was done. How was triangulation utilized? This should be made clear in the study.
Discussion
The discussion is well done.
However, it is a bit confusing when the study was focused on collecting data pertaining to ‘older adults’ but the results and discussion keep talking about impact on ‘families’ with older adults. If the study was about older adults, it gets mixed up when we now look at impact on families. This is done repeatedly throughout the document (impact on older adults/impact on families with older adults).
The framework shows we are looking at families. Was information obtained from any other family members or just restricted to older adults only? How do we generalize their perceptions and experiences to the larger family on the outer spheres of the CSFET? Why was data not collected from any other family members to weave in the different perspectives? The authors can respond to this and give a justification of their approach and the eventual discussions/conclusions.
Conclusions
- There was no section clearly indicated as “Conclusion” section is given. This is important to help the reader understand what are the key take-aways of the study and how this has fitted to the purpose of study.
Limitations
- It is important to explain/list some of the limitations of such a study for the readers and also any other future studies.
- No limitations mentioned yet this is a study that relied mainly on personal memory/recall for responses besides other records used. For the earlier waves the recall may be different than the later/recent waves of the pandemic in Japan.
Overall: A good paper and very interesting topic with relevant and helpful knowledge.
Author Response
Dear Reviewer 2:
Thank you for your comments. In response we have taken the following measures:
Your comment:
-Did younger people/family members play any role in this study? Was the role of younger people considered? Why or why not?
Our answer:
As the study targeted families with older adults, the subjects of the formal interviews and informal interviews were family members including older adults. From line 137 to line 138, this was modified to aid comprehension.
Your comment:
However, the main highlight was on ‘content analysis’. I am not sure that stating “content analysis for this study is a right term’. What was done for this analysis as I understand it is more of “thematic analysis’ and not content analysis’. The authors should check on this and make necessary adjustments.
Our answer:
As text was cited based on reference number 24 on line 193, this is accurately described as directed content analysis. We have clarified this as directed content analysis on line 192.
Your comment:
Rigor. This is explained on how trustworthiness was attained. However, since different sources of information were utilized for data collection, it is not clear how triangulation was done. How was triangulation utilized? This should be made clear in the study.
Our answer:
As is explained on line 156, all data in the field notes were entered qualitatively, so the contents of the field notes were analyzed. To facilitate comprehension, we have added text to reflect this from line 188 to line 189.
Your comment:
Discussion
However, it is a bit confusing when the study was focused on collecting data pertaining to ‘older adults’ but the results and discussion keep talking about impact on ‘families’ with older adults. If the study was about older adults, it gets mixed up when we now look at impact on families. This is done repeatedly throughout the document (impact on older adults/impact on families with older adults).
Our answer:
In order to discuss the impact on families, we have changed older adults to families with older adults from line 372 to line 373.
Your comment:
The framework shows we are looking at families. Was information obtained from any other family members or just restricted to older adults only? How do we generalize their perceptions and experiences to the larger family on the outer spheres of the CSFET? Why was data not collected from any other family members to weave in the different perspectives? The authors can respond to this and give a justification of their approach and the eventual discussions/conclusions.
Our answer:
As indicated previously, the formal interviews and informal interviews targeted all family members including older adults. To aid in comprehension we have explained this in detail from line 137 to line 138. Also from line 465 to line 467, we have added text concerning limitations.
Your comment:
Conclusions
-There was no section clearly indicated as “Conclusion” section is given. This is important to help the reader understand what are the key take-aways of the study and how this has fitted to the purpose of study.
Our answer:
A "Conclusion" section has been added from line 470 to line 484.
Your comment:
Limitations
-It is important to explain/list some of the limitations of such a study for the readers and also any other future studies.
-No limitations mentioned yet this is a study that relied mainly on personal memory/recall for responses besides other records used. For the earlier waves the recall may be different than the later/recent waves of the pandemic in Japan.
Our answer:
"4.3. Limitations and Future Directions" was added from line 463 to line 469.
Reviewer 3 Report
Comments and Suggestions for Authors
Your study is well done and you results are interesting. Please clarify the points I make below.
Who determined the definition/duration of the 7 waves? And why were they not of equal length?
Figures 1 & 2 are very well done and easy to follow. Table 1 is less easy to follow, probably because it has so much information. Can Table 1 be divided into at least two tables, perhaps by putting the positive info in one table and the negative info in another table?
line 153: what are the articles that were collected? Were they published journal articles, or diaries or newspaper articles, etc?
Line 155: what do you mean by statistical materials?
Are your semi-structured interviews with individuals or with groups of individuals?
Author Response
Dear Reviewer 3:
Thank you for your comments. In response we have taken the following measures:
Your comment:
Who determined the definition/duration of the 7 waves? And why were they not of equal length?
Our answer:
Line 101 explains the meaning of the terms, which were obtained from reference number 18. Japan's Ministry of Health, Labour and Welfare determined the duration of the waves based on number of infected persons and other data.
Your comment:
Figures 1 & 2 are very well done and easy to follow. Table 1 is less easy to follow, probably because it has so much information. Can Table 1 be divided into at least two tables, perhaps by putting the positive info in one table and the negative info in another table?
Our answer:
As per your suggestion, from line 217 to line 233, we have separated Table 1 and Table 2.
Your comment:
line 153: what are the articles that were collected? Were they published journal articles, or diaries or newspaper articles, etc?
Our answer:
This study was ethnographic research. Consequently, as you pointed out, analysis was made of any and all obtainable data. This has been newly noted from line 161 to line 162.
Your comment:
Line 155: what do you mean by statistical materials?
Our answer:
The material covered statistics on population, age structure, industry, etc. On line 161, we have revised the term to read statistical data.
Your comment:
Are your semi-structured interviews with individuals or with groups of individuals?
Our answer:
Semi-structured interviews were conducted with a total of 20 family units. This was added on line 163 to line 164.